# Barriers to the Digitization of Health Information: A Qualitative and Quantitative Study in Kenya and Lao PDR Using a Cloud-Based Maternal and Child Registration System

**DOI:** 10.3390/ijerph18126196

**Published:** 2021-06-08

**Authors:** Tarek Numair, Daniel Toshio Harrell, Nguyen Tien Huy, Futoshi Nishimoto, Yvonne Muthiani, Samson Muuo Nzou, Angkhana Lasaphonh, Khomsonerasinh Palama, Tiengkham Pongvongsa, Kazuhiko Moji, Kenji Hirayama, Satoshi Kaneko

**Affiliations:** 1Graduate School of Biomedical Sciences, Nagasaki University, Nagasaki 852-8523, Japan; t_numair@hotmail.com; 2Department of Ecoepidemiology, Institute of Tropical Medicine, Nagasaki University, Nagasaki 852-8523, Japan; daniel.harrell@austin.utexas.edu; 3Dell Medical School, The University of Texas in Austin, Austin, TX 78712, USA; 4School of Tropical Medicine and Global Health, Nagasaki University, Nagasaki 852-8523, Japan; tienhuy@nagasaki-u.ac.jp (N.T.H.); nsmtfts@gmail.com (F.N.); moji-k@nagasaki-u.ac.jp (K.M.); 5Nairobi Research Station, Nagasaki University-Institute of Tropical Medicine—Kenya Medical Research Institute (NUITM-KEMRI) Project, Nairobi 19993-00202, Kenya; yvonne.muthiani@tuni.fi (Y.M.); nzoumuuo@gmail.com (S.M.N.); 6Health Sciences Unit, Faculty of Social Sciences, Tampere University, 33014 Tampere, Finland; 7Centre for Microbiology Research, Kenya Medical Research Institute (KEMRI), Nairobi 54840-00200, Kenya; 8Savannakhet Provincial Health Department, Savannakhet 13000, Laos; tockykpn@gmail.com (A.L.); rasinh.146@gmail.com (K.P.); tiengkhampvs@gmail.com (T.P.); 9Department of Immunogenetics, Institute of Tropical Medicine, Nagasaki University, Nagasaki 852-8523, Japan; hiraken@nagasaki-u.ac.jp

**Keywords:** mother and child health, health record digitalization, antenatal care, postnatal care, cloud-based system

## Abstract

Digitalization of health information can assist patient information management and improve health services even in low middle-income countries. We have implemented a mother and child health registration system in the study areas of Kenya and Lao PDR to evaluate barriers to digitalization. We conducted in-depth interviews with 20 healthcare workers (HCWs) who used the system and analyzed it qualitatively with thematic framework analysis. Quantitatively, we analyzed the quality of recorded data according to missing information by the logistic regression analysis. The qualitative analysis identified six themes related to digitalization: satisfaction with the system, mothers’ resistance, need for training, double work, working environment, and other resources. The quantitative analysis showed that data entry errors improved around 10% to 80% based on odds ratios in subsequent quarters compared to first quarter periods. The number of registration numbers was not significantly related to the data quality, but the motivation, including financial incentives among HCWs, was related to the registration behavior. Considering both analysis results, workload and motivation to maintain high performance were significant obstacles to implementing a digital health system. We recommend enhancing the scope and focus on human needs and satisfaction as a significant factor for digital system durability and sustainability.

## 1. Introduction

Digital medical records are one of the recent technologies used to assist patient information management [1]. The use of digital medical records is also reported to decrease time and cost related to medical information management, which can help governments make up-to-date and evidence-based public health decisions while providing key health indicators toward achieving the United Nations Sustainable Development Goals (SDGs) efficiently and with low cost [2].

Current paper-based documentation of the health records jeopardizes healthcare services with the potential of introducing data errors and causing delays and resulting in the adoption of digital medical records in many healthcare systems. Applying advanced digital technologies can provide real-time accurate information access to healthcare workers (HCWs) and provide decision supports to healthcare professionals for better clinical care provision. Digital registration application also enhances monitoring and interaction among healthcare professionals in different healthcare settings [3]. Electronic system designs for medical purposes must consider feasibility, flexibility, robustness, scalability, and maintenance, which are essential principles to ensure proper system integration to provide better health services [4]. As an example, electronic medical records (EMR) could save more than $81 billion annually. Eventually, as EMR provides proper management of chronic disease and other social problems, it can save up to $142–371 billion per year [5].

Digital medical profiles and electronic health identities provide users of the health system with advantages such as fewer errors, faster work, remote access to data, confidentiality assurance, proper patient follow-up, and proper data backup. In 2014, O’Mahony et al. found that paper registration will always contain data inaccuracies and require more time for recording, resulting in delayed and incomplete registration during periods of high traffic of patients [6]. In contrast, Kang’a et al. studied the implementation of computerized medical records in Kenya and found that after establishing it, exchanging data between records would enhance user’s experience and reduce duplicated patient registration burden [7]. Digitization to record health information should also benefit low and middle-income countries (LMICs). Standardization is necessary for digitization [8,9], but it is difficult for LMICs to standardize health information recording among the entire healthcare system [10].

A maternal and child health (MCH) booklet records antenatal care (ANC) visits, delivery, and postnatal care (PNC) visits for growth monitoring and immunization. The format and information are similar worldwide, which will provide an ideal environment for digital system implementation for LMICs [11]. The digitization of health information in developing countries has begun [12]. In the case of the maternal and child health handbook (MCH handbook), efforts have already started [13,14]. However, there are still barriers to its national-level implementation and dissemination.

We have developed a custom cloud-based MCH registration system, Women and Infant Registration (WIRE), based on the local MCH handbook and have implemented it in health centers in remote areas of Kenya and Lao PDR [15,16]. This research aims to qualitatively and quantitatively identify barriers to digitizing health information such as the WIRE System in resource-limited sub-Saharan African and Southeast Asian clinical settings; specifically, to identify the obstacles associated with the initial implementation and maintenance of the electronic system in remote areas of developing countries and to validate them through registration data.

## 2. Materials and Methods

This study adopted a mixed-method research approach using qualitative and quantitative measures to determine barriers in digitizing health information in Kenya and Lao PDR heuristically [17,18]. According to the methodology applied in previous studies, we heuristically identified the barriers or obstacles for digitalizing health information from qualitative data analyses of in-depth interviews; then, we quantitatively validated the identified issues and detected additional other issues that could not be identified by qualitative analyses using registered data in parallel [19,20,21,22,23].

We set the digitization target as data entry for mother and child registration by entering information contained in the MCH handbook available locally. The WIRE system was used to evaluate the digitalization of health information in this study. Since WIRE utilized a cloud environment, the data-entry persons could enter and access patient’s MCH data via an internet connection. Furthermore, the WIRE implementation did not require an installation of the program on local devices. Further description of the WIRE system is shown in Appendix A.

To identify the barriers and obstructions in the digital registration process and enhance the digitalization of health information in developing countries, we delivered the WIRE system to health facilities in resource-limited regions in Kenya and Lao PDR.

### 2.1. Study Areas

In order to understand generalized trends regarding barriers to digitization, we chose two study areas in Africa and Asia, which share similarities in terms of digitization lags, but have different cultural, environmental, and historical backgrounds. We have delivered the WIRE Systems to register pregnant women, mothers, and infants during routine clinical operations at five health facilities in Kwale County, Kenya and five health facilities in Savannakhet Province, Lao PDR, where the Nagasaki University Institute of Tropical Medicine has had activities for research in tropical medicine and global health [24]. We provided one-day training on the system setup day and another refresher training day one year later for MCH nurses, facility medical records officers, community health workers (CHWs), and clinical officers from each study site.

In Kenya, at the beginning of January 2017, six WIRE systems were delivered in Kwale County at five public sector MCH clinics: two primary care hospitals, one health center, and two dispensaries. Two WIRE systems were delivered at Kinango Hospital because ANC and PNC were operated parallelly with many mothers and children. In 2019, we added four personal computer units and an additional dispensary to improve accessibility, resulting in nine WIRE systems and six research sites. To date, we installed the system into Kwale District Hospital, Kinango Hospital, Diani Health Clinic, Mwaluphamba Dispensary, Mwachinga Dispensary, and Vyongwani Dispensary (Figure 1).

In Lao PDR, at the beginning of August 2017, four WIRE systems were installed at four MCH clinics in Songkhone and Xepon Districts in Savannakhet Province: Songhkone District Hospital, Lahanam Health Center, Xepon District Hospital, and Dongsavanh Health Center. In October 2018, another WIRE system was installed in an additional MCH clinic: Manchy Health center.

### 2.2. Data Collection

#### 2.2.1. Qualitative Data Collection

Qualitative data were collected by an in-depth interview method with 20 HCWs from 10 healthcare settings in both study areas. The participants were nurses, midwives, doctors, community health workers, or medical assistants who had managed the WIRE system and were interviewed by local research team members who were trained at study sites.

A semi-structured interview guide was tested in Kenya with two interviewees to validate the interview guideline. The guideline started with the quest of the questions to establish the background for the interviewees’ work experience, which was followed by open-ended questions to identify the barriers, benefits, and limitations of the WIRE system. Questions were revised according to the pilot interview to include more possible themes to analyze. Interview transcripts were made in the English language; then, they were translated into the local languages by our study team members familiar with English and local languages. In Kenya, the in-depth interviews took place in October 2017 in five health facilities, excluding one dispensary that had not participated at that time. Interviewees included nurses, community health workers, and operators hired exclusively for data entry in the WIRE system. In Lao PDR, interviews were conducted in February 2019 for ten individuals who worked in the WIRE system in the five health facilities. Interviews results were translated by a local Laotian volunteer who is familiar with the English language for further analysis. As no operators were hired in Lao PDR, interviewees included nurses, doctors, and midwives who used the WIRE system on a daily basis. In both Kenya and Lao PDR, the same interview questions were directed to all participants (Appendix A).

#### 2.2.2. Quantitative Data Collection

During the study period, mother and child registration was operated actively for 21 months in Kenya and 31 months in Lao PDR during routine services in study sites following the digital operation guideline [25]. In total, 12,887 mothers and 10,499 children (*n* = 23,386) were registered in the Kenyan study site, and 7132 mothers and 3528 children (*n* = 10,660) were registered in the Laotian study site, respectively, by the end of January 2021. For ANC visits, 6875 and 12,382 visits were registered, while for PNC, 26,624 and 4355 visits were registered in Kenya and Lao PDR, respectively (Appendix A). For quantitative analyses, summary research data were generated from the WIRE system for demographics, ANC, and PNC visits in separate datasets for mothers and infants. Patient demographic information, including a randomly generated WIRE ID number, was separate from their corresponding ANC and PNC medical records. The WIRE ID was used as a cross-walk.

To elucidate the obstacles for digital data entry, we quantitatively analyzed the relationship between registration number and monthly compensation for the enrollment of new mothers and new infants, searching for relative events that might have affected the registration manners for data registrars. We compensated either 50 Kenyan shillings (equivalent to USD 0.50) or 2000 Lao KIP (equivalent to USD 0.25) per entry (i.e., new mother ID).

### 2.3. Data Analysis

#### 2.3.1. Qualitative Analysis

The thematic framework analysis was used to analyze in-depth interviews, where responses from transcripts were coded and analyzed for patterns of themes for responses [26,27,28]. Responses from in-depth interviews were coded and assigned to a pre-existing coding frame, which was structured based on former discussions with HCWs and staff members of our research team before structuring the interview guide (Appendix A). The analysis process started with identifying keywords and thoughts derived from the interviewee’s responses; then, we coded the quote or the text and fitted it appropriately into the coding frame. Analysis results were structured and supported by quotes from the thematic framework. All interviewees were reassigned into numbers to ensure anonymity. Thematic framework analysis was conducted using Microsoft Excel for Mac 2020 by assigning comments to preset themes according to the repetition of responses. Then, the thematic module was structured according to the frequencies of interviewees’ responses.

#### 2.3.2. Quantitative Analysis

After exporting the registration information tables from the WIRE database, data cleaning was conducted to create mother and infant datasets for demographics, ANC visits, and PNC visits. We summed up the presence or absence of the required items for each mother for the ANC dataset and children for the PNC dataset and created outcome variables as dichotomous variables according to whether they were higher (=1) or lower (=0) than the mean value (Appendix A). For the ANC dataset, essential variables used for the analysis were medical staff visit, gestation week, type of service, weight, blood pressure, fundal height, fetus head position, fetus position, fetal heart rate, fetal movement, urine analysis result, paleness, mother condition, fetal condition, and fetal abnormal maturity. Those variables for the infants’ demographics dataset were birthdate, registration date, address, health facility, and sex. The PNC dataset consisted of medical staff conducting visits, age of the infant, temperature, weight, height, type of investigations, and abnormal condition comments.

As independent variables, we created a variable with four levels of quartiles of total enrollment and number of visits during the study period at the medical facility where the mother and child were enrolled, in addition to the mother–child information. In addition, the recorded time was recorded by quarters (three-month unit) and added as a variable for each record. In Kenya, the calculation of quarters began in April 2017 (Q1) and ended in April 2020 (Q13), while in Lao PDR, the calculation began in August 2017 (Q1) and ended in January 2020 (Q10).

Logistic regression analyses were conducted with the dependent and outcome variables to ascertain which variables were related to the leakage of data entry (quality of data entry). The quarter variable and the number of visits were transformed into dummy variables for the analyses.

Furthermore, in order to evaluate whether the presence or absence of incentives affects the number of enrollments, we statistically examined the difference in enrollments between the periods with and without incentives using a Wilcoxon signed-rank test [29].

All statistics were performed using Stata15 (StataCorp LLC. College Station, TX, USA) and presented as values with 95% confidence intervals (CI).

### 2.4. Ethical Consideration

All interviewed HCWs participated voluntarily after being informed of the study purpose, confidentiality rights, and after obtaining verbal approval consent. The research protocol received ethical approval from Kenya Medical Research Institute (KEMRI) in Kenya under KEMRI/SERU/3746, and National Ethics Committee for Health Research (NECHR) in Lao PDR number 2019.25.sav/NECHR.

## 3. Results

### 3.1. Participating HCWs Characteristics

Table 1 shows job titles for interviewed HCWs that participated in in-depth interviews. We interviewed 20 HCW in total with 10 in Lao PDR and 10 in Kenya. Half of the HCWs were midwives in Lao PDR and half were nurses in Kenya. No doctors or midwives participated in the Kenya project, while in Lao PDR, no special operators were hired for the study.

### 3.2. Qualitative Analysis: Semi-Structured In-Depth Interviews’ Outputs

Thematic framework analysis identified several themes related to barriers during the adoption of the WIRE system for mother and child health. Overall, the WIRE system was positively perceived by its users. Many participants’ comments indicated workload issues and how these issues, including internet connectivity, affected the use of the WIRE system in resource-limited clinics (Table 2).

#### 3.2.1. Thematic Framework Analysis

##### Theme 1: HCWs’ Satisfaction with the System

All interviewed users found the WIRE system easy to use either from the start or after days of regular use. A community health worker from Kenya mentioned the need for users who needed time to get used to WIRE system functions:

“At the beginning, we had challenges, but afterward, we become better with time.”

Nine interviewees in Kenya and six in Lao PDR had previous experience with digital software before or at least had a personal computer. They found the data easy to enter without effort for the second time, as a doctor from Lao PDR with a history of having a personal computer said:

“It is easy because the patient’s name was already there; it is not complicated because we just enter data.”

##### Theme 2: Mother’s Resistance

In-depth interviews included a specific question to capture users’ impressions about mothers’ willingness to participate in the study. Kenyan responses expressed more frequent comments relating to the hesitancy of mothers. During our study, Kenya was going through the 2017 national and local elections, resulting in some mothers suspecting that the study’s activities were part of the election process.

“Mothers hesitated with using the system as they were worried that it would be something to do with elections”, a Kenyan nurse mentioned.

The second main reason for mother resistance was related to confidentiality of the health data, especially for HIV patients, as a Kenyan nurse said:

“Confidentiality was the main reason for hesitation. We assured them that the medical data and personal data would be kept confidential. HIV-positive clients hesitated because of privacy concerns. I explained to them the benefit of recording information on the system. Mothers only trust nurses, and once we explained, the mothers become comfortable.”

Concerns from Laotian mothers were focused on the unfamiliarity of the WIRE system, but once they received a brief explanation, concerns were relieved. A doctor from Lao PDR said:

“They would be wondering why health center workers always used mobile phones while asking them. Thus, we had introduced them to understand why we use mobile phones. We will put their data into the mobile phone. It would be comfortable when they are going to district hospital because their information is linked.”

##### Theme 3: Need for Training on the System

The need for additional training on the system was more frequent in Lao PDR, as shown in Table 1. During interviews, the main concern arose from a nurse that had a lack of familiarity with software systems or personal computers. A midwife highlighted this issue as a challenge for low registration:

“Who does not know how to use the system will not do it and will forget to enter the data.”

##### Theme 4: Double Work Perception by HCWs

Most participants reported using the WIRE system in parallel with the regular clinical workflows, resulting in a double work burden. Although users had the feeling of entering the data twice, they still preferred the WIRE system for better accessibility and storing of the data. As a doctor from Lao PDR mentioned:

“Using the WIRE system increased my work after writing on paper, [because] next I saved [it] in [the] WIRE system, then I had to write a report to the district hospital. Using a computer is better because it can save data; the data cannot be lost.”

Other interviewees found the double work burden to be related to the first few days or weeks of using the system, and the work became routine after getting used to the WIRE system with regular paperwork, as a Laotian doctor summed up:

“Yes, it was a workload but not anymore; it is [a] workload because we are not familiar with the system. I like the computer because it is easy and it can increase my computer skills”.

Interviewees in health facilities who receive many clients thought that the extra workload was related to the high flow of patients and could be managed through recruiting more staff, as a Kenyan nurse mentioned:

“When we are busy, it takes [a] long [time] from taking informed consent to clinical observation and then registration on [the] WIRE system; and ANC registration, so it becomes double recording. It added to the workload. It was like an extra job. The workload increased due to the need to enter both paper registers and WIRE systems. We stayed late in the facility. Patients stayed for longer and were delayed. If we were more, it would be easier. More staff to enter data may be required. It was not possible to conduct informed consent for all the mothers.”

##### Theme 5: The Need for a Better Working Environment

Sub-theme a (internet): All interviewed HCWs mentioned factors related to the work environment that they needed to solve or enhance. Internet connection-related issues were the most frequent factor mentioned among interviewees. One of the Laotian doctors mentioned this issue as the only problem as follows:

“The internet signal is low, especially in the rainy season, sometimes no signal all day.”

Sub-themes b (hardware), c (desk and space), and d (power supply): The other most frequent factor was related to the freezing of provided personal computers, lack of adequate space for hardware, power blackout, or lack of colleagues support to reduce work overload, as a nurse in Kenya summed up:

“I did not receive support from my colleagues to recruit and registers mothers. I had to enter the information twice by myself. We wasted a whole week due to power blackouts. At one time, the blackout lasted for two weeks, and the pc battery supply is not sufficient. We had to send clients elsewhere as immunization was also affected. Another challenge was inadequate space. The table was overfilled with many working gadgets, especially [in the] ANC area.”

##### Theme 6: The Need for Resources

Interviewees from Kenya addressed other challenges that they needed to overcome to ensure better performance from their point of view. Some of the comments were related to basic needs for users such as electricity and water, as a WIRE operator in Kenya mentioned:

“[The] electricity socket is far away from the working station; therefore, I had to recharge the laptop in advance or move when the battery on the machine is depleted. Some days, there was no water that resulted in the nurses not effectively doing their work, hence discouraging data entry in the WIRE system.”

Interviewees from study areas mentioned that the need for enhancement of human resources and recruiting more staff is essential to maintain work consistency. One of the Laotian midwives said:

“In my opinion, causes of no data entry are no time to re-enter the data due to many patients, shortage of HCWs and having [to] multi-task, and also a low signal of internet.”

### 3.3. Quantitative Analysis: WIRE System’s Data Output

We summarized the data registered by HCWs to correlate with responses from interviews and verify mentioned factors with registration scores by each quarter.

#### 3.3.1. Observation for Registration Patterns

The registration pattern for ANC and PNC visits for ten quarters from August 2017 to January 2020 in Lao PDR and eight quarters from April 2017 to April 2020 in Kenya (excluding nurses’ strike period) are shown in Figure 2. In Lao PDR, observations from the registration numbers through time showed the disruption of the registration process because of delayed incentives in Q5 and Q7. In Kenya, the national nurses’ strike caused a complete stoppage from Q4 to Q8.

#### 3.3.2. Statistical Correlation to Registration Score

The logistic regression model for registration scores correlation with percentile intervals in monthly registration numbers showed a significantly higher risk for lower scores when a monthly registered number increased. The median interval for registering new clients showed a significant decrease in middle intervals (OR: 0.56 and 0.39 in Lao PDR and Kenya, respectively). Score correlation with the number of visits for mothers showed significance in ANC for Lao PDR datasets for second visits (OR: 0.73). PNC datasets in Kenya showed a high risk for lower scores after the first visit (OR: 0.26 for the second visit and 0.11 for the third visit). Correlation for registration scores with time series in both study areas showed the significant risk for having a below-average score concerning the first quarter of starting the project. For ANC datasets, significance was spotted in different quarters (Q3–Q10 in Lao PDR and Q9–Q13 in Kenya). Correlation for time series in demographic datasets did not show significance in Lao PDR, while it shows the significance of having high scores, up to nine times, in Q10 to Q13 in Kenya: ORs are 2.34, 5.03, 9.1, and 8.42, respectively. PNC datasets in Kenya showed the significance of low scores in Q10, Q11, and Q12: ORs are 0.47, 0.47, and 0.55 (Table 3).

Delayed incentives in Lao PDR during Q5 and Q7 did not result in a significant correlation in the PNC dataset. In Q5, the ANC dataset was significant for having low registration scores (OR: 0.71); while in Q7, the ANC (OR: 0.48) and demographic (OR: 0.31) datasets showed significance for higher risk of having lower scores in comparison to other quarters with regular incentives. The difference in registration numbers between the periods without (Q5 and Q7: *n* = 2) and with an incentive (other quarters: *n* = 8) was statistically significant (*p*-value = 0.04) by the Wilcoxon signed-rank test for the ANC dataset in Lao PDR (Table 3).

## 4. Discussion

In this study, we aimed to identify barriers to digitizing health information qualitatively and quantitatively using the WIRE System, which is a digital MCH registration system developed by us for research purposes. In the qualitative analysis, we categorized the data into six themes by the framework method. We discuss each theme, referring to the results of the quantitative analysis.

The first and third themes were issues on satisfaction with the system and the need for training. Qualitative results in our study showed similarity among study areas for the first theme. The need for training was more frequent in Lao PDR comparing to responses from Kenya. This could be related to hiring special operators for data entry in Kenya, while in Lao PDR, we recruited the working HCWs in health settings for the study. From the interviews in our study, most HCWs were satisfied with the system and gave positive feedback on its usability. They also indicated that their operational maturity increased with time. With the initial training, the data entry skill can be nourished and matured with time, which means that the system itself is not a “barrier” to the digitalization of the health information system, although some training opportunities may be necessary. The result aligns with those of a study in 2014 that indicated that 54.1% of healthcare professionals were ready to use electronic medical records [13].

HCWs, who deal with EMRs or digital systems, tend to be satisfied more with the system’s speed and efficacy. In addition, digital systems are more manageable to use when having computer skills [30]. Similarly, in our study, users with a history of dealing with computers used the WIRE system faster and within a shorter period. The WIRE system was localized in the official national language of study areas to reduce the challenging effect of the language barrier as recommended by Belden et al. in 2009 [31]. The language issue was not mentioned as a barrier or a challenge for users in our study.

The second theme, mother’s resistances in digitalization, was identified from qualitative analysis but was not present in our quantitative one. However, it may become an obstacle for scaling up the digitalization of health information in developing countries. Mother’s resistance would take time to overcome until electronic documentation adoption reaches the point where they cannot imagine medical and nursing care without EMRs as opposed to paper-based records [32]. Furthermore, as developed countries promoted digitalization through policymakers and regulation, LMICs should organize a campaign or program for patients to access educational materials and opportunities to engage in conversations about the risks and benefits of participation in the digital health information system [33]. Our study is focusing on the user’s perspective of possible barriers; still, we considered the feedback from mothers to see if users will face much resistance, which affects their motivation. On the other hand, we captured in interviews the importance of local HCWs attitudes to gain mother’s trust from both study areas. Similar results were found from Egypt in 2014, highlighting the importance of trustful relations between HCWs and patients for better satisfaction of both [34].

The fourth theme was the double work perception for conventional documented registration and electronic registration by this study. This issue was mentioned as the most frequent burden for data entry’s proper performance in both study areas; and it was supported by the high risk of getting a lower score (i.e., poor data entry) when there were many clients above the expected average in Table 3. In our study, the WIRE system did not replace the current routine paperwork system, which resulted in double work for our users. Although many HCWs from in-depth interviews did not find the system an extra burden, most HCWs from both study areas were aware that both workflows, paper-based and digital registration, as a duplication of work. In the future, governments and healthcare systems should mandate and provide incentives toward the mass adoption of electronic documents use to accelerate the digitalization of health information systems in LMICs [35,36,37].

Regarding themes five and six for better working environment and resources, the internet connection issue was a more frequent barrier in Lao PDR; while space and power supply were considered more challenging for the Kenyan HCWs. The need for a better working environment and resources, limited resources, and improvement of the environment at healthcare facilities is a matter for all medical services in LMICs, not only for digitalizing health information [35]. Training nurses specializing in health information or training medical information specialists will be necessary [38].

Additionally, we found that the motivation of data-entry persons became an obstacle for digitalizing health information in LMICs from the quantitative analysis. As shown in Figure 2, incentives for HCWs affected the registration number. Maintaining motivation would directly affect the quality and quantity of registration, according to our results. Furthermore, the number of visits during ANC and PNC visits correlated with having lower scores after the first visit. The close connection between nurses and clients may cause the sensation of less need to fulfill the data. Another explanation could be related to the overload of the registration flow in relation to health service provision leading HCWs to be demotivated [39]. Our study shows common challenges related to double work burden during the pilot phase. The participants in our study expressed high motivation to work with the digital system, especially in the first quarter. Nevertheless, correlation analysis showed good evidence of lower score and performance related to Q2 to Q9 in Lao PDR, and Q2 and Q9 to Q13 in Kenya, indicating a perception of becoming less motivated.

Moreover, in Lao PDR, our project faced fewer logistic challenges in Q5 because of the coordinator change and staff turnover in health facilities, which needed time to reallocate the budget for new members and delayed incentives. Q7 in Lao PDR included the beginning of the new financial year in Japan and the reallocation of funding sources, which caused the second delay in incentives. During this period, the number of registrations reduced, indicating financial incentives, which should be considered to maintain the information systems at health facilities, as Jawhari et al. mentioned [40].

Reduced motivation may be caused by political action that affects the quality and quantity of health record registration. During the implementation of this study, Kenya was affected by the nurses’ strike in early 2017, which affected the delivery of many healthcare services [41,42,43]. The strike was followed by temporary fund suspension resulting in the stopping of activities. Kaguthi et al. recently concluded that investment in Kenya’s health infrastructure and staffing should decrease clients’ congestion at health facilities, improve quality of care, and decline mortality [42]. Statistical analysis from our study did not show the significance of low performance related to financial motivation; still, observation and in-depth interview results highlight the importance of responding to user’s needs and perspectives.

Although the use of the MCH handbook with digital registration was considered as an appropriate incentive for users, providers, and local government in developed countries such as Japan [44], still, financial incentives in parallel with non-financial incentives are crucial for encouraging uptake of EMRs and health digital systems in urban slum areas [40]. Furthermore, lack of financial incentives is considered one of the leading reasons for the current inadequate technologies in sub-Saharan Africa [45]. Kenyan users in our study mentioned financial incentives as a gained benefit from this pilot study, while in Lao PDR, significantly fewer users mentioned the same. Non-financial incentives such as recognition, career development, and strengthening personal qualifications were identified in Africa to be essential for motivating HCWs as a management tool [46].

This study leveraged the MCH handbook registration to evaluate the digital health system’s applicability and sustainability. The original Japanese MCH handbook, developed by the Japan International Cooperation Agency (JICA), is a tool to monitor maternal, newborn, and child health during ANC, delivery, PNC, and vaccination stages, which was introduced in Japan in 1948. Since its initial success in Japan, JICA developed and introduced other MCH books to developing countries in different languages in the 1980s. By the 2010s, JICA-inspired MCH books were used in 25 countries as a part of national MCH monitoring programs [47]. Since the MCH handbook is somehow standardized already, it provides an opportunity to evaluate the feasibility of digitalizing the information system in LMICs, as Williams et al. mentioned [48]. The World Health Organization (WHO) stated that involvement stakeholders and user-testing are essential for developing the ANC toolkit that can be adopted in different health domains [9]. The results of our study showed a need to improve the environment of data entry and patient registration, the solution of incentives, and the avoidance of multiple workloads in the health facilities, in addition to the standardization of information systems.

In terms of the limitation of this study, we only analyzed the quality of the registered data with digitally recorded information, and we could not examine the missing registration. The number of registrations fluctuated, and the cause could not be identified in this study. Concerning the satisfaction level of theme 1 in the qualitative analysis part, this study focused on open-ended interviews and aimed to gain an in-depth understanding of the problems in developing countries. Therefore, we have not conducted a user satisfaction analysis using the structural questionnaires with a Structural Equation Model (SEM) for the digitalizing system in this study [49]. We plan to conduct the HCWs and mother’s satisfaction study following the progress of infrastructure development for digitalization in remote areas in developing countries.

Furthermore, the national nurse strike in Kenya was affected by data collection, which was followed by travel restrictions to both countries caused by the COVID-19 global pandemic. In the future, it is necessary to incorporate user satisfaction surveys as described in the literature to improve the system for promoting digitalization in remote areas of developing countries. Despite those limitations, we could identify the factors to be improved to have a high quality of data for digitizing healthcare information.

## 5. Conclusions

Our study identified barriers to implementing digital systems in resource-limited sub-Saharan African and Southeast Asian clinical settings by adopting a qualitative and quantitative approach using the WIRE system, which is a multilanguage cloud-based MCH registration system. The qualitative analysis identified six themes related to digitalization: HCWs’ satisfaction with the system, mother’s resistance, need for training, double work, working environment, and other resources. The quantitative analysis showed that data entry errors improved around 10% to 80% based on odds ratios in subsequent quarters compared to first quarter periods. The number of registration numbers was not significantly related to the data quality, but the motivation, including financial incentives among HCWs, was related to the registration behavior. Considering both analysis results, workload and motivation to maintain high performance were significant obstacles to implementing a digital health system. We recommend enhancing the scope and focus on human needs and satisfaction to improve motivation as a significant factor for digital system durability and sustainability in the resource-limited sub-Saharan African and Southeast Asian clinical settings. In order to digitize health information in these areas, it is necessary not only to develop electronic systems but also to improve the system and the infrastructure environment.

## Figures and Tables

**Figure 1 ijerph-18-06196-f001:**
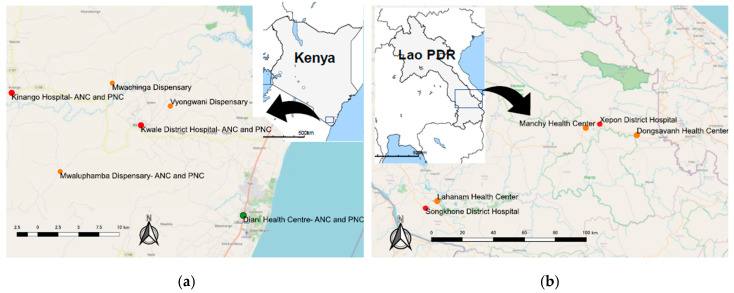
Study sites and health facility locations: (**a**) Kwale county in Kenya: Kwale District Hospital, Kinango Hospital, Diani Health Clinic, Mwaluphamba Dispensary, Mwachinga Dispensary, and Vyongwani Dispensary; (**b**) Savannakhet province in Lao PDR: Songhkone District Hospital, Lahanam Health Center, Xepon District Hospital, Dongsavanh Health Center, Manchy Health Center.

**Figure 2 ijerph-18-06196-f002:**
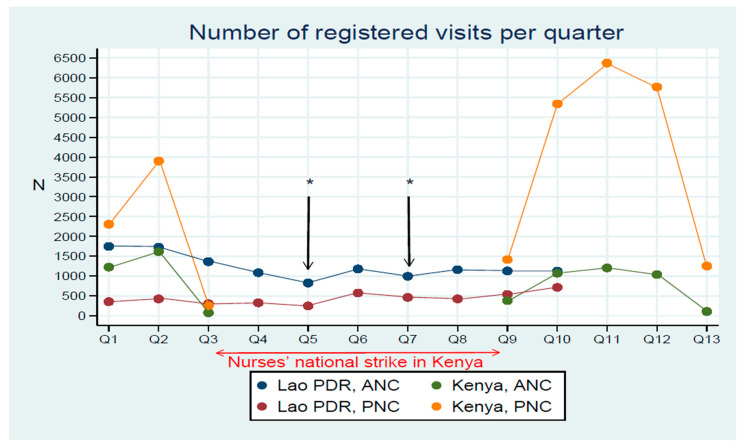
Numbers of registration at antenatal care (ANC) visits and postnatal care (PNC) visits per quarter in two study areas. In Lao PDR, there were delayed financial incentives in Q5 and Q7 (*). In Kenya, there was no registration process during Q4–Q8 because of the national nurses’ strike. Quarters in Kenya: from April 2017 (Q1) until April 2020 (Q13); those in Lao PDR: from August 2017 (Q1) until January 2020 (Q10).

**Table 1 ijerph-18-06196-t001:** Job titles of healthcare workers interviewed after working in the WIRE system.

Job Titles	Kenya	Lao PDR	Total
Nurses	5	1	6
Midwife	0	5	5
Doctors	0	2	2
Community health worker	2	0	2
Medical assistant in primary healthcare	0	2	2
Operator (recruited for the study)	3	0	3

**Table 2 ijerph-18-06196-t002:** Summary of comments for themes and sub-themes.

Themes/Sub-Themes		Number of Comments	
		Lao PDR	Kenya
Theme 1: Satisfaction with the system *		15	8
Theme 2: Mother’s resistance		3	6
Theme 3: Need for training on the system		11	1
Theme 4: Double work perception *		20	15
Theme 5: Need for a better working environment *	Theme 5a: Internet	18	4
	Theme 5b: Hardware	1	9
	Theme 5c: Desk and space	--	4
	Theme 5d: Power supply	--	2
Theme 6: Need for resources		5	6
Total comments		73	55

*: Most frequent themes resulted from in-depth interviews.

**Table 3 ijerph-18-06196-t003:** Results of multivariate logistic regression analysis to percentile intervals in monthly registration numbers and number of visits analyses * to evaluate data entry quality in both study areas, Lao PDR, and Kenya.

	Lao PDR						Kenya					
Variables	ANC ^a^ (*n* = 12,382)		Demographics ^b^ (*n* = 10,660)		PNC ^c^ (*n* = 4355)		ANC ^a^ (*n*= 6875)		Demographics ^b^ (*n* = 23,386)		PNC ^c^ (*n* = 26,624)	
	OR ^d^	95% CI ^e^	OR ^d^	95% CI ^e^	OR ^d^	95% CI ^e^	OR ^d^	95% CI ^e^	OR ^d^	95% CI ^e^	OR ^d^	**95% CI ^e^**
**Percentile of monthly registration number in each area ^f^**												
**25th**	Reference		Reference		Reference		Reference		Reference		Reference	
**50th**	0.73	0.66–0.81	0.56	0.37–0.86	1.07	0.61–1.88	0.89	0.76–1.03	0.39	0.31–0.49	1.31	1.08–1.57
**75th**	0.76	0.68–0.84	0.66	0.43–1.02	0.57	0.35–0.92	1.37	1.18–1.6	0.24	0.19–0.29	1.27	1.05–1.53
**100th**	1.69	1.51–1.89	1.02	0.62–1.65	0.86	0.52–1.44	1.97	1.73–2.25	0.22	0.18–0.27	1.34	1.12–1.59
**Number of visit records ^h^**												
**One visit**	Reference		NA ^g^		Reference		Reference		Reference		Reference	
**Two visits**	0.73	0.54–0.98			0.13	0.06–0.28	0.98	0.61–1.59	NA ^f^		0.26	0.18–0.37
**Three visits**	0.31	0.07–1.4			0.14	0.02–1.27	1.14	0.42–3.07			0.11	0.03–0.39
**Four visits or more**	0.42	0.16–1.11			No visits		2.05	0.65–6.48			No visits	
**Time series (quarters)**												
**Q1**	Reference		Reference		Reference		Reference		Reference		Reference	
**Q2**	0.91	0.77–1.07	1.09	0.52–2.29	0.89	0.37–2.16	0.33	0.28–0.38	1.77	1.49–2.1	1.05	0.76–1.45
**Q3**	0.52	0.4–0.62	1.04	0.47–2.29	0.76	0.31–1.91	16.1	3.92–65.79	7.99	1.96–32.55	3.33	0.81–13.72
**Q4**	0.56	0.47–0.67	0.71	0.36–1.41	0.58	0.25–1.37	NA ^f^ (Due to nurse striking in Kenya)					
**Q5**	0.71	0.58–0.87	0.85	0.38–1.88	2.15	0.58–8.03						
**Q6**	0.56	0.47–0.67	0.64	0.34–1.18	1.65	0.65–4.22						
**Q7**	0.48	0.4–0.57	0.31	0.17–0.54	1.08	0.44–2.64						
**Q8**	0.43	0.36–0.51	0.23	0.13–0.39	1.22	0.48–3.1						
**Q9**	0.55	0.46–0.66	0.59	0.29–1.17	0.86	0.38–1.96	0.31	0.24–0.39	0.96	0.82–1.13	0.47	0.34–0.67
**Q10**	0.19	0.17–0.23	1.71	0.63–4.57	0.44	0.21–0.92	0.21	0.17–0.25	2.34	1.95–2.83	0.47	0.35–0.63
**Q11**	NA ^f^ (Due to end of registration program in Lao PDR)						0.13	0.11–0.16	5.03	3.8–6.66	0.55	0.41–0.73
**Q12**							0.16	0.13–0.19	9.1	5.78–14.32	1.16	0.85–1.58
**Q13**							0.18	0.11–0.27	8.42	2.67–26.48	3.67	1.82–7.42

*: We summed up the presence or absence of the required items listed below for each dataset and created outcome variables as dichotomous variables according to whether they were higher (=1) or lower (=0) than the mean value.; ^a^: for the antenatal care (ANC), essential variables were medical staff conducting visit, gestation week, type of service, weight, blood pressure, fundal height, fetus head position, fetus position, fetal heart rate, fetal movement, urine analysis result, paleness, mother condition, fetal condition, and fetal abnormal-maturity; ^b^: for the demographics, those were birthdate, date of registration, address, health facility, and sex for infants; ^c^: for postnatal care, those were medical staff conducting visits, age of the infant, temperature, weight, height, type of investigations, and abnormal condition comments.; ^d^: odds ratio; ^e^: 95% confidence interval; ^f^: Monthly registration number: categorical variable based on the 25% percentile interval; ^h^: The number of visits’ record: categorical variable based on the total number of visits for each client; ^g^: not applicable.

## Data Availability

The data that support the findings of this study are available from the corresponding author, S.K., upon reasonable request.

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
