# Peer review of "Barriers to the Digitization of Health Information: A Qualitative and Quantitative Study in Kenya and Lao PDR Using a Cloud-Based Maternal and Child Registration System"

_ijerph, 2021, doi:10.3390/ijerph18126196_

Round 1
Reviewer 1 Report
- In a qualitative and quantitative analysis there should be a concrete methodological framework that will retain the research conducted within a specific framework. The authors should ask specific research questions or make particular assumptions that may or may not be justified by the results.
- When someone targets to study the users satisfaction from an information system, he/she should consider specific methodological frameworks as the ones discussed in "A Literature Review of User Satisfaction Models Towards Information System Success", Antonopoulou, M., & Kotsilieris, T. (2019), International Journal of E-Services and Mobile Applications (IJESMA), 11(2), 71-87. doi:10.4018/IJESMA.2019040105. If the authors deliberately ignored these frameworks they should justify why
- In page 5 line 218 you state: “Half of the HCWs were nurses and midwives in Kenya and Lao PDR, respectively. No 218doctors or midwives participated in the Kenya project” It is unclear to the reader what is the number of midwives. One has to read the table in order to understand. Please revise this sentence.
- In section 3.2.1 you pose some claims that are not justified by numbers or data provided.
- 3.3.1 There should be a capital letter in the beginning of the section title.
- The results’ discussion is rather descriptive than in depth. It provides no insight to the reader.
- The conclusion section has to be revised so as to support its purpose that is to mention the objectives and summarize the findings of the paper.
Author Response
We appreciate the valuable comments. The purpose of this research is to understand and analyze qualitatively the impressions and opinions of people who used digital input systems for the first time in the remote areas of Africa and Asia, which are far from the digital world, and to understand the problems of digitization of health and medical information in the remote areas of developing countries by analyzing the actual input data quantitatively. This research aims to identify the obstacles to digitizing health information in remote areas of developing countries. Although it may be necessary to conduct a study focusing on user satisfaction when digitalization becomes acceptable and implementable, we do not have this objective in this study. Considering the purpose of this study, we would like to respond to the reviewer's questions and comments and revise the manuscript accordingly.
- In a qualitative and quantitative analysis there should be a concrete methodological framework that will retain the research conducted within a specific framework. The authors should ask specific research questions or make particular assumptions that may or may not be justified by the results.
Response 1:
We agree with this valuable comment. To clarify the points mentioned at the beginning of this response letter, we have added the following wording to the purpose of the introduction, as the reviewer indicated (Lines 84-86) as below.
"To identify the obstacles associated with the initial implementation and maintenance of the electronic system in remote areas of developing countries and to validate them through registration data."
- When someone targets to study the users satisfaction from an information system, he/she should consider specific methodological frameworks as the ones discussed in "A Literature Review of User Satisfaction Models Towards Information System Success", Antonopoulou, M., & Kotsilieris, T. (2019), International Journal of E-Services and Mobile Applications (IJESMA), 11(2), 71-87. doi:10.4018/IJESMA.2019040105. If the authors deliberately ignored these frameworks they should justify why
Response 2:
Thank you for your comment and for sharing the valuable reference by M.Antonopoulou et al. (2019).
As reviewer 1 pointed out, user satisfaction is essential at some stage. However, as we mentioned initially, we wanted to identify the problems or obstacles through this study, the situation of the medical field leading to the implementation of digitization, and its potential. The research results also concluded that it is necessary to improve the system as a medical environment infrastructure rather than the electronic system itself, such as infrastructure improvement, double-work elimination, and motivation maintenance.
Although we recognize that the examination of user satisfaction as described by reviewer 1 is a necessary survey, we think it to be conducted in the next stage when the infrastructural situation of the medical environment in developing countries is improved and the operation of the digital system reaches a stage where it can be performed stably.
Instead of the methodology in the reference the reviewer shown, we adopted open-ended interviewees to achieve that, and we validated responses from interviews in the quantitative analysis section, a mixed-methods, to inductively identify the barriers to digitalization. This method was adopted in other similar published researches (references 19-23). We have added a more detailed justification paragraph in the method section (L88-94), as recommended, with newly cited contents as examples.
- In page 5 line 218 you state: "Half of the HCWs were nurses and midwives in Kenya and Lao PDR, respectively. No 218doctors or midwives participated in the Kenya project" It is unclear to the reader what is the number of midwives. One has to read the table in order to understand. Please revise this sentence.
Response 3:
We thank the kindest comment. We have revised and rephrased the sentence to be more understandable in the revised manuscript.
- In section 3.2.1 you pose some claims that are not justified by numbers or data provided.
Response 4:
We appreciate the kind comment for improving our manuscript, and we added more information in the result section 3.2.1. We also added more explanation and rephrasing in themes 1 and 2. We added one more quotation in theme 2 to justify the results more. Also, to meet the kind comment, we have discussed these results more in the discussion section in the revised manuscript.
- 3.1 There should be a capital letter in the beginning of the section title.
Response 5:
The beginning of section titles was capitalized as recommended.
- The results' discussion is rather descriptive than in depth. It provides no insight to the reader.
Response 6:
We thank the kind suggestion. We added a more detailed discussion for a better debate of the results in the discussion section.
- The conclusion section has to be revised so as to support its purpose that is to mention the objectives and summarize the findings of the paper.
Response 7:
We thank the kind suggestion. The conclusion section was revised following the reviewer's comment.

Reviewer 2 Report
The article addresses a very pertinent subject that still needs more attention from the Social Sciences, Medical Sciences and others. In order to strengthen the text, I leave some comments:
1-In the introduction, the aim of the article could be further developed / detailed / problematized. Only the following is said: “This research aims qualitatively and quantitatively to identify 82 barriers to digitizing health information using the WIRE System in resource-limited sub-83 Saharan African and Southeast Asian clinical settings”.
2-In the presentation of the methodology (articulation of the quantitative and qualitative approaches) it would be pertinent to justify this type of approach.
3-Perhaps it is justified to identify, in some way (maintaining anonymity), the interviewees at the end of the excerpts used in the results section.
4-The analysis lacks a theoretical-conceptual discussion around the central subject: Barriers to the digitization of health information. Therefore, two possibilities are suggested: (a) create another section before the methodological section to make a brief assessment of the state of the art; OR (b) in the discussion section, make an even more detailed debate of the results obtained, systematically considering other scientific investigations on similar subjects.
5-Suggestion: develop the conclusion a little further and take the opportunity to indicate here possible paths for the future development of the research.
Author Response
We appreciate the valuable comments. We have addressed all the suggestions raised by the reviewer2 and revised the manuscript accordingly.
- In the introduction, the aim of the article could be further developed / detailed / problematized. Only the following is said: "This research aims qualitatively and quantitatively to identify 82 barriers to digitizing health information using the WIRE System in resource-limited sub-83 Saharan African and Southeast Asian clinical settings".
Response 1:
We thank the kind comment. We have received other valuable comments from another reviewer to make the research aim more concrete and specific. We did our best to comprise all comments, and we developed the article's aim to meet all the comments. We made our research questions specific to identify barriers to implementing a digital medical system from the user's point of view in health facilities as below in Lines 84-86.
"To identify the obstacles associated with the initial implementation and maintenance of the electronic system in remote areas of developing countries and to validate them through registration data."
- In the presentation of the methodology (articulation of the quantitative and qualitative approaches) it would be pertinent to justify this type of approach.
Response 2:
The guidelines for mixed-method research by Judith Schoonenboom and R. Burke Johnson (Citation 18 in the main context) illustrated many dimensions of designs for mixed-methods and at the same time stated that "designs should follow from one's research questions and purposes, rather than questions and purposes following from a few currently named designs." We, therefore, designed methodological frameworks that best serve our research questions. The research aims to capture as many barriers as possible. We adopted open-ended interviewees to achieve that, and we validated responses from interviews in the quantitative analysis section, a mixed-methods, to inductively identify the barriers to digitalization. This method was adopted in other similar published researches (citations 19-23). We have added a justification paragraph in the methods section (Line 88-94), as recommended, with newly cited contents as examples.
- Perhaps it is justified to identify, in some way (maintaining anonymity), the interviewees at the end of the excerpts used in the results section.
Response 3:
Our concern in the research regarding the anonymity of interviewees was because of the limited sample size, which could jeopardize the anonymity of interviewees. We appreciate the kind suggestion for improving our manuscript, and we added more information in the result section 3.2.1 after and before each quotation. We also added more explanation and rephrasing in themes 1 and 2. Also, we added one more quotation in theme 2 to justify the results more.
- The analysis lacks a theoretical-conceptual discussion around the central subject: Barriers to the digitization of health information. Therefore, two possibilities are suggested: (a) create another section before the methodological section to make a brief assessment of the state of the art; OR (b) in the discussion section, make an even more detailed debate of the results obtained, systematically considering other scientific investigations on similar subjects.
Response 4:
We thank the valuable suggestion. As we received a similar comment from another review report for the discussion part, we choose option (b). Accordingly, the discussion part was more detailed to meet all reviewers' satisfaction. We added a more detailed discussion for a better debate of the results.
- Suggestion: develop the conclusion a little further and take the opportunity to indicate here possible paths for the future development of the research.
Response 5:
We appreciate the kindest suggestion. We revised the conclusion section following the reviewer's comment.

Round 2
Reviewer 1 Report
I insist that the issue of Satisfaction must be tackled as it is Theme 1 of the work.
Figure 2 must be updated as it is not clear.
You should further argue about the reasons that lie behind your choice of these two countries (i.e. what do you expect to find out by this comparison?). The fact that the University has medical infrastructures in both countries is not enough. Also, Discussion section should further focus on a comparison of the findings among Kenya and Lao PDR.
There are several typos and grammar errors that have to be revised. Several words should start with a capital letter and plural is used instead singural and vice versa.
Author Response
Response to Reviewer 1 Comments
- I insist that the issue of Satisfaction must be tackled as it is Theme 1 of the work.
We added this issue to our study's limitation in the discussion part, citing the recommended paper by Reviewer 1 in the previous review comments (cited as reference 49 in the main text). The added part in the discussion (L502-508) is as below:
Concerning the satisfaction level of Theme 1 in the qualitative analysis part, this study focused on open-ended interviews and aimed to gain an in-depth understanding of the problems in developing countries. Therefore, we have not conducted a user satisfaction analysis using the structural questionnaires with Structural Equation Model (SEM) for the digitalizing system in this study [49]. We plan to conduct the HCWs and mother’s satisfaction study following the progress of infrastructure development for digitalization in remote areas in developing countries.
- Figure 2 must be updated as it is not clear.
We used figure 2 to discuss the observed obstacles to digitalization in unstable conditions in developing countries. We added an explanation in the results section 3.3.1 (L341-343). We updated the figure and the legend of the figure as recommended. Also, we used Figure 2 for discussion that the incentives affected the behavior for registration in Line 444- 446.
- You should further argue about the reasons that lie behind your choice of these two countries (i.e. what do you expect to find out by this comparison?). The fact that the University has medical infrastructures in both countries is not enough. Also, Discussion section should further focus on a comparison of the findings among Kenya and Lao PDR.
We thank the suggestion from reviewer 1. We added the following description about the reason why we chose the two countries for this study in the methods section (L109-111) about the reason why we used two areas in Africa and Asia as below:
" In order to understand generalized trends regarding barriers to digitization, we chose two study areas in Africa and Asia, which share similarities in terms of digitization lags, but have different cultural, environmental, and historical backgrounds. "
We addressed differences between the two areas in discussion when relevant or when statistical significance was shown from the results. Following the valuable suggestion from reviewer1, we have added more statements in the discussion section under each theme to highlight more similarities and differences between the two study areas in Lines 393-396, Lines 420-425, Line428 and Lines 437-439.
- There are several typos and grammar errors that have to be revised. Several words should start with a capital letter and plural is used instead singural and vice versa.
We reconfirmed and corrected the errors by a native English co-author as suggested.

Round 3
Reviewer 1 Report
It seems fine at the present form